# Parents' views of psychological research with children: Barriers, benefits, personality, and psychopathology

**Stefanie M. Jungmann** \*, **Galyna Grebinyk, Michael Witthöft**

Department of Clinical Psychology, Psychotherapy, and Experimental Psychopathology, Johannes Gutenberg-University Mainz, Mainz, Germany

\* jungmann@uni-mainz.de

**Data Availability Statement:** The original data is available at: DOI: 10.17605/OSF.IO/JVFNA.

**Funding:** The authors received no specific funding for this work.

## Abstract

Psychological studies with children have difficulty recruiting participants and samples are more often selective. Given parental consent for children's participation, this study examined parents' perceived barriers and benefits of participating in studies and associated parental personality and psychopathological characteristics. Since there are hardly any instruments available so far, the study also aimed to develop questionnaires for the systematic and standardized assessment of barriers and benefits. One hundred and nine parents with children < 18 years completed questionnaires on willingness to participate, perceived barriers (Parents' Barriers for Participating in Research Questionnaire, P-BARQ) and benefits (Parents' Benefits for Participating in Research Questionnaire, P-BERQ), personality traits, trait anxiety, and psychopathological characteristics. The P-BARQ and P-BERQ showed overall acceptable model fits (TLI/CFI = .90–.94; RMSEA = .08/.14) and internal consistencies (α = .68–.86). Parents' willingness to own participation in psychological studies and their support for children's participation correlated negatively with perceived barriers to participation ($r \geq |\text{-.32}|$, $p < .001$). Parental personality traits (such as agreeableness/ openness) showed positive associations with one's own participation ($r \geq .19$, $p < .005$) and negative correlations with perceived barriers to participation ($r \geq |\text{-.24}|$, $p < .001$), while parental psychopathological characteristics are more closely related to consent to children's participation ($r = .24$, $p < .05$). Parental trait anxiety showed both a positive correlation with perceived barriers (uncertainty) and benefits (diagnostics/help) ($r \geq .20$, $p < .05$). For the willingness to participate in studies, barriers seem to play a more crucial role than the benefits of participation. If more information is given about psychological studies, uncertainties and prejudices can be reduced.

## Introduction

The participation of children and adolescents in psychological studies usually depends on parental consent. From a legal and ethical point of view, the participation of children and adolescents in studies requires the informed consent of their guardians or legally authorized

**Competing interests:** The authors have declared
that no competing interests exist.

representatives [1–3]. This is an important regulation for the protection of children and adolescents, which is also enshrined in regulations in Europe and the United States [4, 5].

Compared to adult studies, the factors that motivate participation in studies involving children and adolescents are much more complex and clearly influenced by parental/family factors, especially in younger children [6–9]. Previous studies have found, for example, correlations between study consent and socio-demographic variables (e.g., higher socioeconomic status), personality traits (e.g., extraversion), and psychopathological characteristics of parents (e.g., no/less alcohol/drugs, in some studies depressive symptoms) [10–14]. At the same time, studies with children/adolescents and families show comparatively low response rates (30–60%) [1, 15, e.g., particularly low at 30–35% in German and US-American prevention programmes, 16, 17] and high dropout rates up to 60% [10, 18–20]. Moilanen et al. [21] found a parental consent rate of 30–40% to a sexuality-related study among adolescents and Pérez et al. [11] found in an epidemiological mental health longitudinal study that between 42–54% of children and adolescents participated in the entire study. If the sample composition is influenced by parental/family factors such as those mentioned above, this could change the study results and could lead to significant limitations [7, e.g., no representativeness, small samples and limited statistical power, reduced reliability and validity; 10, 21]. Especially in the context of child and adolescent research, it seems essential to investigate which factors contribute to parental consent or rejection/termination to reduce study limitations and promote recruitment and retention [7, 14, 21].

Previous studies have found that European participants or being White, higher education of parents and higher socio-economic status are overrepresented in child and adolescent research, while cultural minorities and participants with risk behaviors (e.g., substance use) are underrepresented [22–25]. An epidemiological study on mental health of children and adolescents (survey of children/adolescents and parents) showed that rejection of participation was more likely for families with a lower socioeconomic status, parental unemployment, cultural minorities, and children with lower school performance [11]. Other studies found that parental consent was more likely in younger and physically ill children [8, 15, 26]. With regard to risk behaviors, studies showed that adolescents with and without active written parental consent significantly differ in substance use, while adolescents with parental consent consumed less tobacco, alcohol, and marijuana [27, 28]. The meta-analysis by Liu et al. [1] on the effects of parental consent in studies on juvenile risk behavior found that, compared to passive consent (consent if not disagreed), active consent overrepresented female and younger participants and underrepresented African Americans. Regarding participating in parental interventions, a systematic review of 28 randomized controlled trials on, in particular, parent training (e.g., on prevention on behavioral problems) identified a younger age, a lower level of education, and a lower socioeconomic status of parents, and belonging to a cultural minority as barriers to participation, although the results were not consistent across the trials. Additionally, parental depression was assessed, which also showed no consistent correlation with willingness to participate [7]. Overall, however, there is a research gap regarding (parental) factors for willingness to participate in more general psychological studies (basic research), especially among infants and school children from the general population (as opposed to specific topics and target groups such as e.g., children with physical illnesses).

In the context of study participation, theories of decision-making and prediction of behavior are relevant. The Prospect Theory [29] assumes that people choose the behavior in which the subjectively expected benefit combined (multiplied) with the expected probability of this positive benefit is the highest. Heuristics may play a role in this process [30]; in the case of study participation, for example, the availability heuristic (e.g., memories of study advertisements or previous study participation). Because study participation is in a social and societal context, altruistic behavior (i.e., selfless behavior) is also assumed in study participation, which

in explanatory approaches may be influenced by several factors, such as situational factors (e.g., mood, time pressure). A reference to health behaviors is provided by the Health Belief Model [31], which assumes that behavior depends in particular on a cost-benefit trade-off, i.e., on assumed advantages and disadvantages of the behavior. Thus, parents would be assumed to agree to and participate in a study if the benefits are perceived to be high and the barriers are perceived to be low [21, 32].

In addition to the above mentioned participant characteristics, beliefs and perceptions of the costs and benefits of participation were also examined. Participation in studies is more likely if the participants trust the research institution, are interested in scientific research, and evaluate a participation personally or socially as beneficial [12, 21, 33]. Vanhelst et al. [8] found that regardless of the child's state of health, the frequent reasons for participation were: a direct benefit for one's own child, altruistic motives, and a low risk of harm to one's own child. A systematic literature review on clinical drug research (38 studies) found similar parental participation factors: personal benefit to the child, altruistic motivation, confidence in safety, and the relation with the researcher [34]. Parents tend to assume that research may be harmful to their own children (e.g., overwhelming children by inquiring about certain topics such as sexuality; e.g., in clinical asthma research, parents considered discomfort to be more strongly linked to harm/risk than the participating adolescents themselves), so that parents want to protect their children's well-being by refusing to participate in a study [21, 35]. This is also shown to be dependent on certain personality traits. For example, Moilanen [12] found that parents with a higher degree of extraversion were more likely to agree to their adolescents participating in sexuality-related research. Regarding more general aspects of studies, Pérez et al. [11] identified individual feedback and shorter duration of the study as relevant aspects for participating in research on children.

## The current study

These effects of parental consent may lead to lower response rates and limit the reliability and validity of studies with children [1, 21, 36]. Previous studies have focused almost exclusively on specific subgroups and special topics (e.g., chronically ill children and adolescents, foster children, migration background, adolescent risk behavior, or drug use; [1, 6, 19, 21, 34, 35, 37, 38]) or on the willingness to participate in intervention or prevention programs [10, 13, 16–18]. In contrast to studies on these specific samples and research areas, the majority of child and adolescent psychology studies examine children and adolescents from the general population and use methods that are hardly expected to cause harm (e.g., no invasive techniques or sensitive issues). However, as the above-mentioned epidemiological studies, for example, show, there are problems in recruiting sufficiently large samples. Overall, there is little research on the links between parents' willingness to participate in psychological studies with children [e.g., 18, 39], parental personality traits [e.g., 12] and psychopathological characteristics [e.g., 13, 18]. For example, parents with children between 5–9 years of age showed a higher dropout rate, while parents with higher extroversion are more likely to agree to participate [7, 12]. With regard to psychopathological characteristics, inconsistent findings are found, particularly with regard to depression, so Robinson et al. [7] conclude in their review that further studies are necessary. In addition, there are hardly any standardized questionnaires that can specifically assess parents' perceived barriers and benefits of participating in psychological studies with children and adolescents. Standardized measurement instruments have the advantages of, for example, providing a more unified and systematic survey, which might have a positive impact on test quality criteria (and allow for optimization), and facilitate application across different samples and countries and thus the comparability of results.

The aims and hypotheses of this study were: (1) The development of standardized self-report measures for assessing parents' perceived barriers and benefits of psychological studies with children. Two questionnaires (benefits/barriers) were developed based on the above empirical findings (e.g., short duration, individual feedback as perceived benefits) and on a similar questionnaire on barriers regarding own study participation from the adult population [42] and were validated in the present pilot study. Given that these were new developments, we expected at least satisfactory psychometric qualities (in terms of reliability, factorial, convergent, and discriminant validity). (2) To investigate the relationship between parents' willingness to participate and perceived barriers and benefits on the one hand and the relationship of these factors with parental personality traits, trait anxiety, psychopathological burden, and depression on the other. In addition to replicating the associations between willingness to participate and benefits (e.g., individual feedback, shorter time duration), uncertainty/mistrust is expected to be most associated with a parental refusal to participate. Regarding personality traits and psychopathology, it was hypothesized that parents higher in extraversion, openness, and agreeableness are willing to participate and parents with higher trait anxiety and psychopathological burden, and depression tend to not be willing to participate. Due to the research gap, the focus should be on studies in infants and school children (recruitment kindergarten and primary school). In Germany, the factors hindering and promoting willingness to participate and related variables are less studied than in other countries (e.g., the USA), but they are also an important research topic in Germany, as shown by the partly low response rates mentioned above [17, 40]. (3) By examining the advantages and barriers that parents consider relevant, as well as the links with personality traits and psychopathology, implications for practice/future studies are to be derived [34].

## Materials and methods

### Participants

This online study was addressed to parents of at least one child under 18 years of age (regarding children no further inclusion or exclusion criteria). Further inclusion criteria were an age of the participants (here parents) of at least 18 years and informed consent. This study required active consent (on the background in Germany: from an ethical and data protection point of view, active consent is required e.g., in online studies by ticking a respective box). A power analysis showed an adequate sample size to be $N = 88$ (G*Power, $\rho = .30$, $\alpha = 0.05$, Power (1-$\beta$) = .90) for correlations and a minimum limit for Structural Equation Modeling (SEM) is recommended to be $N = 100–150$ [41, 42], including 10% drop-outs, the target size was at least $N = 110$.

One hundred and twelve parents (i.e., 112 different families) took part in the study. Three parents reported having children under 18 years of age, but the stated age of the children was over 18 years, so these three participants were excluded from the data analysis. The final sample included 109 participants. The average age was $M = 32.4$ years ($SD = 5.7$, range 21–54) and 87% of the participants were female. Table 1 shows the sociodemographic characteristics of the parents and their families.

### Design and procedure

The present study was a cross-sectional online survey to pilot the newly developed questionnaires and to investigate the above-mentioned relationships. Recruitment for the survey titled "What do parents think about psychological studies with children" took place via a primary school and a kindergarten in a large German city as well as via social media (Facebook, Instagram, and Telegram) and parents' forums (e.g., on leisure or parenting). At school and

**Table 1. Sociodemographic characteristics of parents and their families.**

| | N | % | M | SD |
|---|---|---|---|---|
| Age (parents) in years | | | 32.4 | 5.7 |
| Sex (parents) female | 95 | 87.2 | | |
| Marital status | | | | |
| Single | 6 | 5.5 | | |
| Relationship | 31 | 28.4 | | |
| Married | 68 | 62.4 | | |
| separated/divorced | 4 | 3.7 | | |
| Widowed | 0 | 0 | | |
| Education (% higher education) | 70 | 64.2 | | |
| Occupation | | | | |
| Unemployed | 1 | 0.9 | | |
| in training | 1 | 0.9 | | |
| Student | 17 | 15.6 | | |
| employee or civil servant | 44 | 40.4 | | |
| self-employed occupation | 6 | 5.5 | | |
| housewife/-husband | 7 | 6.4 | | |
| on parental leave | 32 | 29.4 | | |
| Others | 1 | 0.9 | | |
| Family income (per month) | | | | |
| < 900 € | 1 | 0.9 | | |
| 900 - < 1300€ | 11 | 10.1 | | |
| 1300 - < 1500€ | 5 | 4.6 | | |
| 1500 - < 2000€ | 8 | 7.3 | | |
| 2000 - < 2600€ | 17 | 15.6 | | |
| 2600 - < 3600€ | 12 | 11.0 | | |
| 2600 - < 5000€ | 38 | 34.9 | | |
| ≥ 5000€ | 15 | 13.8 | | |
| no information | 2 | 1.8 | | |
| Number children (per family) | | | 169 (1.5) | (0.7) |
| Age (children) | | | 3.2 | 3.6 |
| < 1 year | 34 | 20.1 | | |
| 1–3 years | 73 | 43.2 | | |
| 4–5 years | 23 | 13.6 | | |
| 6–11 years | 30 | 17.8 | | |
| 12–14 years | 3 | 1.8 | | |
| 15–17 years | 2 | 1.2 | | |
| (reported siblings ≥ 18 years) | 4 | 2.4 | | |
| Sex (children) female | 57 | 33.7 | | |

*Note.* N = 109.

kindergarten, parents received an e-mail with study information (type, content, and duration) and the link to the study (i.e., no other persons such as gatekeepers were involved in the recruitment). Recruitment in social media and on platforms for parents included a post with the same study information and the link to the study. Before participants could begin the online survey, they gave informed consent after receiving detailed study information (i.e., checking boxes indicating that information was read and understood and that they willingly

participated in the study as opposed to the option of not read, not understood, or no voluntary participation). All participants then initially completed sociodemographic information and the questionnaires listed under Measures. Participants did not receive any expense allowance for the online survey. The present study was approved (2018-JGU-psychEK-021) by the local ethics committee of the Department of Psychology.

## Measures

**Parents' Barriers for Participating in Research—Questionnaire (P-BARQ).** Previous studies have found as barriers or reasons for dropouts: a long study duration, lack of feedback, and assumptions that participation in the study could be burdening/harmful for the children [11, 21]. In addition, the development of the items was based on the subscales of the Barriers to Research Participation Questionnaire [BRPQ; 43, 44] which includes the subscales mistrust, religious beliefs, health beliefs/fears, role overload/time demands, and incentives (in this study related to own participation in adults, not that of children). On the basis of the above findings and the BRPQ, 15 items were formulated (e.g., "I cannot find time to participate.", "I am thinking about what negative effects participation could have for my child.", "I do not want others to know personal information about our family."; all items see S1 Table). The agreement to the statements is evaluated on a five-point Likert Scale from 1 = "strong rejection" to 5 = "strong agreement".

**Parents' Benefits for Participating in Research–Questionnaire (P-BERQ).** Previous studies have found that aspects such as perceived personal advantage, altruistic motives, individual feedback, relationship/personal contact with the investigator, and shorter duration are factors relevant to parents that increase willingness to participate [8, 11, 12, 21, 33, 34]. Based on the specific aspects mentioned above that parents consider to be beneficial, 14 items were developed for this study (e.g., ". . . if a diagnostic interview is conducted and I receive feedback about the mental health of my child.", ". . . if the duration is as short as possible, i.e. less than one hour."; ". . . a telephone call with the head of studies or investigator is made before participation."; all items see S2 Table). The items are rated on a five-point Likert Scale from 1 = "strong rejection" to 5 = "strong agreement".

**Big Five Inventory-short version (BFI-10).** The short version of the Big Five Inventory [45, 46] comprises 10 items covering the five dimensions of personality (2 items each): openness to experience, conscientiousness, extraversion, agreeableness, and neuroticism (five-point Likert scale from 1 to 5). The BFI-10 showed acceptable structural, convergent, and external validity [47, 48]. Based on the aim of having two items per scale represent the respective facet as broadly as possible, Rammstedt et al. [45] argue that the internal consistency of the two items per scale is unlikely to be an adequate measure of reliability. Previous studies found an average retest reliability (six weeks) between $r_{tt}$ = .56 and $r_{tt}$ = .73 [47, 48].

**State-Trait Anxiety Inventory, Trait version (STAI-T).** The trait version of the State-Trait-Anxiety Inventory [47] measures anxiety as a personality trait using 20 statements with a four-point Likert scale (0 = "not at all" to 3 = "severely"). The trait version showed an excellent internal consistency ($\alpha$ = .90) and high correlations with other anxiety measures ($r$ = .73 $-r$ = .90) [47]. In this study, the internal consistency was $\alpha$ = .92.

## Symptom Checklist-Short (SCL-S)

The SCL-S (Bleichhardt & Hiller, unpublished) is a short form of the Brief Symptom Inventory [48], which assesses the general psychological burden with 15 questions. The items are rated on a four-point Likert scale with regard to their occurrence in the last seven days (1 = "a little" to 4 = "very strong"). The sum score of the SCL-S showed a high correlation with the Global

Severity Index (GSI) of the BSI ($r$ = .95; Bleichhardt & Hiller, unpublished). In addition, a sub-scale for the severity of depression with 6 items can be calculated [49], which showed high reliability ($\alpha$ = .87) and satisfactory factorial, convergent and discriminant validity [50]. In this study, the internal consistencies for the general psychological burden/total SCL-S ($\alpha$ = .92) and for the subscale depression ($\alpha$ = .90) were excellent.

### Previous study participation and current willingness to participate

The parents were asked how often they and their children have taken part in psychological studies so far. The current willingness to participate was asked with three items: for their own participation, to support the child's participation, and to assess whether their child would agree to participate in a study (1 = "definitely not participate" to 5 = "definitely participate"; see S3 Table). In addition, different titles of studies were given and the parents were asked to assess their willingness to participate in the study with the mentioned title on a non-verbal scale (smiley, 1–5). Based on common titles, we varied the term of the study context (3 versions: 'research project', 'study of the University XY', 'scientific study') as well as the term for the target group of children and adolescents (2 versions: 'children and adolescents as participants', 'young participants aged between XX and XY years'), resulting in 6 different titles (e.g., 'Study of the University XY looks for children and adolescents as participants, see all six titles S4 Table).

### Statistical analyses

The statistical analyses were performed using the software programs SPSS Statistics 23 [51] and Mplus 7 [52]. For the P-BARQ and the P-BERQ, the initial number of factors was determined by using parallel analysis [53] including principal components analysis for normally distributed random data generation. Subsequently, Explorative Structural Equation Models (ESEM) [54, 55] were calculated. The mean and variance-adjusted weighted least square algorithm (WLSMV) was used as model estimator. For the evaluation of the model quality, the absolute fit index RMSEA (Root Mean Square Error of Approximation) and the incremental fit indices CFI (Comparative-Fit-Index) and TLI (Tucker-Lewis-Index) were used. According to Hu and Bentler [56], RMSEA values < .08 and CFI as well as TLI values > .90 indicate an acceptable model fit. Pearson correlation coefficients were calculated to test the relationships to the second research question. In addition, $t$-tests were conducted to examine comparisons between children and parents in terms of estimated current willingness to participate and number of previous study participations. Since it was not possible to skip questions in the online study, there were no missing values in the data set. We checked for duplicates and response patterns (one person responded the same to each of the items in the P-BARQ or P-BERQ, but no person responded the same across all questionnaires, so no exclusion occurred). The original data is available at: DOI: 10.17605/OSF.IO/JVFNA.

## Results

### Participants' characteristics

Table 2 shows the mean values, standard deviations, and Cronbach's $\alpha$ values of the used self-reports (total and subscales). The parents rated their own willingness to participate in a psychological study significantly higher compared to the willingness of the child ($t$(108) = 7.97, $p$ < .001, $d$ = 0.88). Parents and children did not differ significantly in the number of participations in previous psychological studies ($t$(108) = 1.96, $p$ =. 053, $d$ = 0.26), with parents showing a very wide range in responses (range: 0–200, $M$ = 4.13, $SD$ = 21.60; most frequent answer with

**Table 2. Participants' characteristics regarding the used questionnaires with mean values (M), standard deviations (SD) and Cronbach's alpha (α) values.**

| Items/Questionnaires | M | SD | α |
|---|---|---|---|
| Previous own study participation | 4.13 | 21.60 | [1] |
| Previous child's study participation | .09 | .87 | [1] |
| Current willingness to own participation (1–5) | 3.55 | 0.91 | [1] |
| Current willingness support child's participation (1–5) | 2.91 | 1.10 | [1] |
| Estimation of the child's willingness (1–5) | 2.66 | 1.10 | [1] |
| P-BARQ total (1–5) | 2.98 | 0.58 | .80 |
| P-BARQ time (1–5) | 2.94 | 0.75 | .68 |
| P-BARQ uncertainty/interest (1–5) | 2.99 | 0.63 | .77 |
| P-BERQ total (1–5) | 3.73 | 0.57 | .83 |
| P-BERQ organizat./study specific (1–5) | 3.84 | 0.57 | .78 |
| P-BERQ diagnostic/help (1–5) | 3.52 | 0.91 | .86 |
| BFI openness (1–5) | 3.65 | 0.91 | [2] |
| BFI conscientiousness (1–5) | 3.60 | 0.70 | [2] |
| BFI extraversion (1–5) | 3.44 | 0.91 | [2] |
| BFI agreeableness (1–5) | 3.22 | 0.81 | [2] |
| BFI neuroticism (1–5) | 2.91 | 0.88 | [2] |
| STAI-T (0–3) | 1.99 | 0.51 | .92 |
| SCL-S total (1–4) | 1.95 | 0.77 | .92 |
| SCL-S depression (1–4) | 1.69 | 0.88 | .75 |

*Note.* N = 109. P-BARQ = Parents' Barriers for Participating in Research–Questionnaire, P-BERQ = Parents' Benefits for Participating in Research–Questionnaire, BFI = Big Five Inventory, STAI-T = State-Trait Anxiety Inventory, SCL-S = Symptom Checklist-Short. [1] each one item, [2] Rammstedt et al. [45] stated that the internal consistency is not suitable as an estimator of reliability due to the low number of items and the intended heterogeneity of the items

67% no previous study participation). Considering the item level at P-BARQ (see S1 Table), the following barriers were associated with low willingness to participate: 'I have little knowledge about current scientific research projects in which my child could participate' (M = 4.04, SD = 1.11), 'I am unsure what to expect when we participate' (M = 3.25, SD = 0.95), and 'I do not want others to know personal information about our family' (M = 3.25, SD = 1.29). In the P-BERQ (see S2 Table), the following benefits were associated with being more willing to participate: '. . . is explained, how my child's participation could help other children" (M = 4.28, SD = 0.80), '. . . I/we would receive feedback on our individual results' (M = 4.11, SD = 0.92), and '. . . the participation takes place exclusively online/at home' (M = 4.06, SD = 0.95).

With regard to the title of the study, parents indicated the highest willingness to participate in the study for the following study title: 'Study of the University XY seeks young participants aged between xx and xx years' (M = 3.45, SD = 1.22; scale: 1–5; see S4 Table).

## P-BARQ and P-BERQ: Factor structure, reliability, and associations with sociodemographic variables

For the P-BARQ, the parallel analysis showed two empirical eigenvalues above the 95% percentile of randomly generated eigenvalues, indicating a two-factor solution. The ESEM with two factors indicated an acceptable model fit: $\chi2(105) = 953.47$, $p < .001$, CFI = .94, TLI = .91, RMSEA = .08 [90% CI: .06 –.11]. A subscale with 11 items represents the factor "uncertainty/lack of interest", the second subscale with 4 items the factor "time" (factor loadings see S1 Table). For similar factor loadings or double loadings (items 1 and 15), the content fit was

**Table 3. Pearson correlations between study participation, perceived barriers and benefits, personality traits, and psychopathological characteristics.**

| | 1 | 2 | 3 | 4 | 5 | 6 | 7 | 8 | 9 | 10 | 11 | 12 | 13 | 14 | 15 | 16 | 17 | 18 | 19 |
|---|---|---|---|---|---|---|---|---|---|---|---|---|---|---|---|---|---|---|---|
| *Previous participation* | | | | | | | | | | | | | | | | | | | |
| 1. own | | | | | | | | | | | | | | | | | | | |
| 2. child | .12 | | | | | | | | | | | | | | | | | | |
| *Current willingness participating* | | | | | | | | | | | | | | | | | | | |
| 3. own | .15 | .07 | | | | | | | | | | | | | | | | | |
| 4. support child | .15 | .02 | .57*** | | | | | | | | | | | | | | | | |
| 5. estimation of child | .03 | -.05 | .34*** | .66*** | | | | | | | | | | | | | | | |
| *Barriers* | | | | | | | | | | | | | | | | | | | |
| 6. P-BARQ total | .01 | -.06 | -.49*** | -.43*** | -.24* | | | | | | | | | | | | | | |
| 7. P-BARQ time | -.08 | .08 | -.42*** | -.32*** | -.16 | .69*** | | | | | | | | | | | | | |
| 8. P-BARQ uncert./ interest | .05 | -.11 | -.44*** | -.40*** | -.23* | .95*** | .44*** | | | | | | | | | | | | |
| *Benefits* | | | | | | | | | | | | | | | | | | | |
| 9. P-BERQ total | -.02 | .02 | .15 | .17 | .14 | .01 | .11 | -.03 | | | | | | | | | | | |
| 10. P-BERQ study specific | .04 | .04 | .15 | .16 | .09 | -.03 | .03 | -.05 | .84*** | | | | | | | | | | |
| 11. P-BERQ diagn./ help | -.08 | -.01 | .09 | .11 | .14 | .05 | .15 | < .01 | .79*** | .33*** | | | | | | | | | |
| *Personality/ Psychopathology* | | | | | | | | | | | | | | | | | | | |
| 12. BFI extraversion | -.06 | .07 | .17 | .02 | .03 | -.19 | .02 | -.24* | .07 | .09 | .03 | | | | | | | | |
| 13. BFI agreeableness | -.02 | -.04 | .19* | .10 | .07 | -.36*** | -.23* | -.35*** | -.19 | -.15 | -.15 | .21* | | | | | | | |
| 14. BFI conscientiousness | -.15 | .11 | -.07 | -.21* | -.18 | < .01 | -.02 | .01 | -.08 | -.03 | -.10 | .09 | .13 | | | | | | |
| 15. BFI neuroticism | .14 | .07 | .08 | .12 | .05 | .03 | -.10 | .08 | .08 | .06 | .07 | -.38*** | -.28** | -.16 | | | | | |
| 16. BFI openness | .01 | .16 | .20* | .09 | .15 | -.15 | -.03 | -.18 | -.03 | -.02 | -.02 | .10 | .10 | .01 | -.06 | | | | |
| 17. STAI-T | .14 | < .01 | .09 | .16 | .05 | .17 | < .01 | .21* | .16 | .07 | .20* | -.44*** | -.33*** | -.22* | .50*** | -.09 | | | |
| 18. SCL-S total | .21* | .02 | .14 | .24* | .15 | .07 | -.06 | .12 | .11 | .02 | .16 | -.38*** | -.21* | -.11 | .38*** | -.08 | .80*** | | |
| 19. SCL-S depression | .17 | < .01 | .08 | .15 | .12 | .10 | -.05 | .14 | .07 | -.05 | .17 | -.40*** | -.15 | -.08 | .35*** | -.08 | .78*** | .94*** | |

*Note.* N = 109. P-BARQ = Parents' Barriers for Participating in Research–Questionnaire, P-BERQ = Parents' Benefits for Participating in Research–Questionnaire, BFI = Big Five Inventory, STAI-T = State-Trait Anxiety Inventory, SCL-S = Symptom Checklist-Short. *p < .05, **p < .01, ***p < .001.

decisive for the assignment to the factor. The Cronbach's α values varied between α = .68 and α = .80 (see Table 2). The item-total correlations were between $r_{it}$ = .31 and $r_{it}$ = .74 (see S1 Table), the inter-correlation of the two subscales was r = .44 (p < .001, see Table 3).

For the P-BERQ, the parallel analysis also identified two empirical eigenvalues above the 95% percentile of randomly generated eigenvalues. Using ESEM, the model fit was: $\chi 2(64)$ = 191.16, p < .001, CFI = .93, TLI = .90, RMSEA = .14 [90% CI: .11 –.16]. There was a subscale "organizational aspects/study-specific benefits" with 9 items and a subscale "feedback and help on psychological problems" with 5 items. In cases of double loadings (items 10 and 11), the content fit was decisive for the assignment to the factor (see S2 Table). The Cronbach's α values varied between α = .78 and α = .86 (see Table 2). The item-total correlations were between

$r_{it}$ = .30 and $r_{it}$ = .77 (see S2 Table), the correlation between the two subscales was $r$ = .33 ($p <$ .001, see Table 3).

Considering sociodemographic variables, the parents' age showed no significant correlations with the willingness to participate, perceived barriers or benefits ($r \leq |.15|$, $p \geq .12$). The children's age correlated negatively with the willingness to own study participation ($r$ = -.23, $p$ = .019) and perceived benefits (P-BERQ subscale organ./study specific: $r$ = -.20, $p$ = .041), and positively with perceived barriers (P-BARQ total: $r$ = .25, $p$ = .010; subscale uncertainty/lack of interest: $r$ = .26, $p$ = .007). Parental education showed significant negative relationships with perceived barriers (P-BARQ total: $r$ = -.20, $p$ = .039; P-BARQ subscale uncertainty/lack of interest: $r$ = -.21, $p$ = .029). Family income showed no significant correlations with the variables studied ($r \leq |.19|$, $p \geq .053$).

## Relationships between willingness to participate, personality traits, and psychopathological characteristics

The willingness to participate in studies and to support the participation of their children showed significant negative correlations with the perceived barriers to study participation, the highest correlation between the parental willingness to own participation and the total value of the P-BARQ ($r$ = -.49, $p <$ .001; see Table 3). At item level, the strongest correlation was with the item "I don't want my child or me to disclose personal information" ($r$ = -.46, $p <$ .001). Considering the validity of the P-BARQ, the significant medium-strong negative correlations with the willingness to participate in psychological studies indicate the convergent validity ($r$ = -.23 –-.49, $p \leq .019$), no significant correlations with perceived benefits of psychological studies may indicate the discriminant validity. In addition, in terms of discriminant validity, no significant associations of the P-BARQ with psychopathology was found, except a positive association between the subscale uncertainty of the P-BARQ and trait anxiety ($r \leq .21$, $p \geq$ .029).

Regarding the relationships between barriers and personality, the P-BARQ (especially the uncertainty/lack of interest subscale) showed significantly negative relationships with extraversion ($r$ = -.24, $p$ = .012) and agreeableness ($r$ = -.23 –-.36, $p \leq .01$).

In contrast to the P-BARQ, the P-BERQ showed no significant correlations with the willingness to participate in studies or to support the participation of their children ($r \leq .17$, $p \geq$ .079). At item level, there were some significant correlations: the highest value was found for the correlation between the willingness to participate and the feedback of individual study results ($r$ = .27, $p$ = .004). The subscale diagnostic feedback and help of the P-BERQ showed a significantly positive correlation with trait anxiety ($r$ = .20, $p$ = .039; see Table 3).

As expected, agreeableness ($r$ = .19, $p$ = .047) and openness ($r$ = .20, $p$ = .042) correlated significantly positively with the parental willingness to the own participation. These personality traits showed no significant correlation with the parents' willingness to support the child's participation ($r \leq .10$, $p \geq$ .323). Parental support for the child's participation was significantly positively associated with the overall parental psychopathological burden ($r$ = .24, $p$ = .013), but there was no significant association between parental willingness for their own participation and the overall psychopathological burden ($r$ = .14, $p$ = .15; see Table 3).

## Discussion

The aims of the present study were (1) to develop and test standardized questionnaires to identify barriers and benefits of psychological studies in children from the parents' point of view, (2) to examine relationships between parent's willingness to participate, barriers and benefits as well as parental personality and psychopathological traits, and (3) to derive implications for

practice/future studies. These aims seem relevant to promote (international) comparability of the investigation and results of the parents' perceived barriers and benefits for psychological studies, to reduce potential study limitations, and to promote improvements for the recruitment/retention of participants [7, 10, 21].

### P-BARQ and P-BERQ as newly developed questionnaires

The P-BARQ and the P-BERQ showed satisfactory model fits and acceptable to high internal consistencies (α = .68 –.86; hypothesis 1). Only the RMSEA value of .14 for the P-BERQ was outside the acceptable range. Although the RMSEA value is usually robust against the sample size, the value may be unacceptable for very small samples or in combination with other factors [57]. For example, redundancy among items can also explain an increased RMSEA value. Some P-BERQ items show high intercorrelations (up to $r$ = .86, child mental health feedback items) and high double loadings (items 10 and 11). For two reasons we have not excluded items with lower factor loadings or double loadings: 1. All items showed acceptable item-total correlations ($r_{it} \geq$ .30), indicating a satisfactory discriminative power of the items [58]. 2. This is a first examination of the questionnaires with a comparatively small sample for ESEM analyses, so an exclusion would be recommended if our results will be confirmed on a larger sample.

For the P-BARQ, support for convergent (medium to strong correlations with willingness to participate) and discriminant validity (missing correlations with benefits and psychopathology, except trait anxiety) could be observed (hypothesis 1). In contrast to the P-BARQ, the P-BERQ showed no significant correlations with the willingness to participate. This could indicate that the perception of barriers is more important for the willingness to participate than perceived benefits. In line with the study by Pérez et al. [11], that individual feedback of study results is seen as a key advantage, at item level we found the highest correlation between willingness to participate and the corresponding item (individual feedback of study results; $r$ = .27, $p$ = .004).

### Willingness to study participation and sociodemographic data

With regard to sociodemographic variables, we could confirm previous studies [8, 26]. Our study extends previous research by providing results on specific factors of barriers and benefits. The older the children are, the more uncertainty and lack of interest were indicated as barriers ($r$ = .26, $p$ = .007). It is possible that parents perceive a lower degree of controllability in older children/adolescents (e.g., children/adolescents report personal information about the family), which may lead to uncertainty. With increasing age of the children, the parents estimated organizational aspects and feedback on the individual test results to be less important ($r$ = -.20, $p$ = .041). This is probably also due to increasing autonomy in adolescence (e.g., adolescents coming alone to the test appointment). In addition, parents with a higher level of education indicated less uncertainty ($r$ = -. 21, $p$ = .029), possibly they are more familiar with research from their own educational background (usually university studies).

### Willingness to study participation and parental personality traits and psychopathology

In line with our hypotheses, we found that agreeableness and openness are positively correlated with the willingness to own participation and negatively related to perceived barriers of study participation. The results could indicate that personality traits (agreeableness and openness) are more relevant for one's own participation in studies and that psychological stress might be more linked with parents' consent to the child's participation.

The direction of the correlations between parental support for the child's participation in a psychological study and parental psychopathology was unexpected. The higher the parental overall psychopathology ($r = .24$, $p = .013$) and trait anxiety ($r = .16$, $p = .09$), the more likely they would support the children's participation. This could be explained by the following related reasons: As our results showed, parents with higher psychological stress have so far participated more frequently in psychological studies themselves ($r = .21$, $p = .030$), i.e., they are more familiar with psychological studies, perhaps the barrier revealing personal information might be lower (e.g., also due to previous psychological support). In addition, we found a significant correlation between trait anxiety and the desire to receive a psychological diagnostic investigation and help through participation in psychological studies. Parents with increased psychological stress may, on the one hand, generally have an increased need for psychological counselling. On the other hand, children of parents with psychological stress and mental disorders have an increased risk suffering from psychological stress and mental disorders [59, 60], which might also result in a desire for diagnosis and support related with the child's participation.

## Implications and future studies

Our study suggests that perceived barriers may play a greater role in the willingness to participate in studies than benefits. With regard to barriers, increased information and empowerment of parents seems to be essential in order to reduce or remove existing uncertainties as barriers. On the one hand, this might refer to the offer and possibilities of psychological studies in a more general way, on the other hand to the type and the detailed procedure of the concrete study. This is also in line with the ratings on the title and recruitment of the participants. Parents rated information about the scientific nature and background of the study as important [33]. Information can also help to clarify misunderstandings regarding the disclosure of personal information about one's own family (e.g., exactly what information is requested, confidentiality). Compared to other countries (e.g., USA), Germany is more reserved and less public in advertising studies and recruiting participants (e.g., less often in public transport or public places). This study showed that in this sample in Germany the most frequently cited barrier to participation in a study is the lack of knowledge about current studies, while at the same time there is a relatively high willingness to participate in principle. This points to the need for increased advertising and information for parents about studies that are taking place. For example, several research institutions could form a cooperation in order to jointly and clearly publish current opportunities to participate in studies. This study also showed further helpful aspects for recruitment (e.g., this study title was rated as particularly positive: "Study of the University XY seeks young participants aged between xx and xx years"). O'Lonergan and Foster-Harwood [61] also showed, for example, that an audiovisual presentation of study information can increase the understanding of research with children/adolescents, both among children and parents. With regard to the motivation and advantages of studies, an altruistic motive was first mentioned, which is in line with previous research [8, 34]. In addition, as previous studies have shown [11], parents consider individual feedback to be relevant. Since especially parents with increased anxiety and psychopathological stress could hope for psychological diagnosis and help through participation, it is relevant to explicitly state this point in the study information in order to avoid misunderstandings and to refer the parents to appropriate services. Parents rated their own willingness to participate higher than that of their children. Further studies are required to assess whether this statement is linked with the protection of children [21] or whether children actually have a lower willingness to participate.

## Limitations

The sample size is clearly at lower bound for the computation of ESEM models [41, 42] and a replication in larger samples is definitely needed. This can also explain the increased RMSEA values. One possible reason that recruitment was more difficult in this study (smaller sample size), despite great efforts, could be explained by the title 'What do parents think about psychological studies with children' and content, as it is more general and less personally relevant than other content-related topics (e.g., COVID-19 pandemic). Overall, the results of the newly developed standardized questionnaires should be interpreted as preliminary (in the sense of a pilot study), further studies with a larger sample are necessary. For a future validation study with a larger sample, it would also be worthwhile to compute an entire structural equation model (using confirmatory factor analysis) of the included constructs and their relationships in order to a apply a more rigorous statistical test on the proposed model and to obtain reliable estimations of the respective latent correlations between constructs (in a subsequent step to examine links with personality and psychopathology). This approach also offers the possibility to compare models and their quality (e.g., with vs. without control of variables or mediator variables). It would also be important for future studies to examine test-retest correlation to adequately assess reliability. Some items of the P-BARQ and P-BERQ show a redundancy and the P-BARQ total score represents almost exclusively the subscale uncertainty/lack of interest. This should also be examined in further studies. The results of this study refer to parents who are probably willing in principle to participate in psychological studies (as these parents WERE willing to participate in this study). Thus, the results must be interpreted against this background, and the sample studied is likely to be different from parents who are opposed to such studies in principle and did not participate (e.g., these parents may perceive other benefits and barriers, or they may show a lower/higher level of anxiety or psychopathology). Therefore, it would be very important for future research to ask specifically these parents, e.g., by explicitly addressing these parents or, in case of rejection of participation, asking them to answer some questions about reasons for rejection. Although the range of previous parental participation was very large (0–200 previous study participation; $M = 4.13$, $SD = 21.60$), most parents had not previously participated in any psychological study (67% no previous study participation), so it can be speculated that possibly both parents who are more open-minded and more skeptical about studies participated in our survey. In addition, the method of recruitment may have influenced the sample characteristics (i.e., not representative) and possibly the results. Although all participants were contacted digitally (e.g., e-mail, social media, forums) with the same information, the specific type of recruitment (via an institution or in a private setting) could probably produce differences in the sample variables and results (e.g., age, education, willingness to participate or correlations with willingness). The specific type of recruitment (or how participants heard about the study) was not recorded as a variable, so no conclusions can be drawn about possible differences due to the specific type of recruitment. Our sample is not representative in terms of socio-demographic data. The low proportion of male participants (13% male and 87% female) is not unusual for parent studies but could lead to biases in the overall results. Some assumptions or few previous findings indicate that fathers might, for example, pay more attention to the organizational aspects (e.g., because of mostly full-time jobs) or show more frequent refusals/dropouts [62, 63]. The parents ($M = 32.4$, $SD = 5.7$) and their children ($M = 3.3$, $SD = 3.6$) were comparatively young, which could also be due to the type of recruitment (i.e., in kindergarten/primary school parents with younger children). As indicated by the employment status, many mothers were on parental leave, during which they presumably found more time to participate (compared to mothers who are employed). Also, the type of recruitment (digital/online) may have attracted parents with an affinity for the

Internet, who may also be more willing to participate in online studies (or psychological studies in general). Even if it can be argued that parents with younger children are parents who may be recruited for future studies in the field of child and adolescent psychology, the results must be interpreted against this background, i.e., parental consent for participation of kindergarten and primary school children in psychological studies. In this context, an item of the P-BARQ ('My child has no interest in scientific research.') presupposes children's knowledge of what scientific research is (especially child-oriented education) so that parents can assess the corresponding interest. Since this knowledge was not recorded and cannot be assumed for children under 3 years of age, the item should be interpreted cautiously. It is also important to mention that ethnicity was not surveyed in this study. As previous studies have also identified cultural minority as one of the barriers to willingness/participation in studies [7, e.g., 11], this is an interesting variable. It would be important to investigate and understand the background of the barriers in different ethnicities (especially ethnic minorities) in more detail, and also to get more representative samples in order to clarify possible misunderstandings and to create equal opportunities/access to participate in e.g., prevention programs/feedback in studies. Our own experiences from recruitment in previous studies suggests that, e.g., parents with lower socio-economic status and/or belonging to a cultural minority were more often mistrustful in the recruitment process (e.g., they suspected that studies were collaborating with public agencies such as the youth welfare office). Although this is consistent with previous studies [e.g., 44], we have no direct empirical evidence on this aspect for our current study.

## Conclusions

Initial evidence suggests that the newly developed questionnaires, the P-BARQ and the P-BERQ, might represent economic, reliable, and valid instruments for assessing parents' perceived barriers and benefits of participating in psychological studies. The perceived barriers seem to be more relevant for the decision to participate in studies than the perceived benefits. While personality traits such as agreeableness and openness were associated with one's own participation and perceived barriers, trait anxiety and psychopathological characteristics seem to be more associated with the willingness to support children's study participation and perceived benefits of study participation (especially diagnosis and help). Additionally, the findings show that parents are willing to support psychological studies if they are sufficiently informed. To ensure and promote the willingness of parents to participate in psychological studies with children and adolescents, it seems particularly important to reduce the lack of knowledge about opportunities to participate in studies, uncertainties, and misunderstandings as barriers. This could be achieved, for example, through scientific communication about psychological studies in general and more information about specific studies (e.g., cooperation between institutes for recruitment, so that information about different participation opportunities can be summarized). Since individual feedback and altruistic motivation were mentioned in particular with regard to benefits, it would be favorable, for example, in research on physical and mental health to consider aspects such as promoting health for the family itself or other children. In addition, current life events and conditions seem to play a relevant role as shown by the large numbers of participants in studies related to the COVID-19 pandemic [64].

## Supporting information

**S1 Table. Parents' Barriers for Participating in Research—Questionnaire (P-BARQ): Items, factor loadings, mean values (*M*, scale: 1–5), standard deviations (*SD*), and item-total correlations (*r*$_{it}$).** *Note. N* = 109. Italic font indicates corresponding factor. [1]all

correlations $p \leq .001$.
(DOCX)

**S2 Table. Parents' Benefits for Participating in Research—Questionnaire (P-BERQ): Items, factor loadings, mean values (*M*; scale 1–5), standard deviations (*SD*), and item-total correlations (r_it).** *Note*. *N* = 109. Italic font indicates corresponding factor. 1all correlations $p \leq .001$.
(DOCX)

**S3 Table. Items on previous study participation and willingness to participation with mean values (*M*) and standard deviations (*SD*).** *Note*. *N* = 109.
(DOCX)

**S4 Table. Six different study titles with mean values (*M*) and standard deviations (*SD*) (willingness participating, scale: 1–5).** *Note*. *N* = 109.
(DOCX)

## Author Contributions

**Conceptualization:** Stefanie M. Jungmann.

**Data curation:** Stefanie M. Jungmann.

**Formal analysis:** Stefanie M. Jungmann, Galyna Grebinyk.

**Investigation:** Galyna Grebinyk.

**Methodology:** Stefanie M. Jungmann.

**Project administration:** Stefanie M. Jungmann.

**Supervision:** Stefanie M. Jungmann, Michael Witthöft.

**Writing – original draft:** Stefanie M. Jungmann.

**Writing – review & editing:** Stefanie M. Jungmann, Galyna Grebinyk, Michael Witthöft.

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
