## [Decision Letter · Decision Letter 0]

20 Mar 2023

PONE-D-23-02403Parents’ views of psychological research with children: Barriers, benefits, personality, and psychopathologyPLOS ONE

Dear Dr. Jungmann,

Thank you for submitting your manuscript to PLOS ONE. After careful consideration, we feel that it has merit but does not fully meet PLOS ONE’s publication criteria as it currently stands. Therefore, we invite you to submit a revised version of the manuscript that addresses the points raised during the review process.

In order for the manuscript to be accepted all the comments by both reviewers need to be addressed fully. The comments related with the methodology (the issue of the small sample, the statistical power analysis), as well as the comments pertaining to the theoretical background of the study, seem to be especially relevant. In line with reviewers' comments, I recommend authors to explicitly integrate their research more with previous relevant research and theory in the introduction section (e.g., citing more relevant literature on this topic). In its current version, I do not think this is done satisfactory.

We look forward to receiving your revised manuscript.

Kind regards,

Srebrenka Letina, Ph.D.

Academic Editor

PLOS ONE

3. We note that you have referenced (Bleichhardt G, Hiller W. Short Version of the Symptom Checklist (SCL-S): Unpublished manuscript. Outpatient Clinic for Psychotherapy, Johannes Gutenberg-University Mainz; 2007.) which has currently not yet been accepted for publication. Please remove this from your References and amend this to state in the body of your manuscript: (ie “Bewick et al. [Unpublished]”) as detailed online in our guide for authors

Additional Editor Comments:

Both reviewers suggest Major Revision of your submitted paper.

The revision of the manuscript has to address all of the comments raised by the reviews.

Reviewers' comments:

Reviewer's Responses to Questions

**Comments to the Author**

1. Is the manuscript technically sound, and do the data support the conclusions?

Reviewer #1: Yes

Reviewer #2: Yes

2. Has the statistical analysis been performed appropriately and rigorously? 

Reviewer #1: Yes

Reviewer #2: Yes

3. Have the authors made all data underlying the findings in their manuscript fully available?

Reviewer #1: Yes

Reviewer #2: No

4. Is the manuscript presented in an intelligible fashion and written in standard English?

Reviewer #1: Yes

Reviewer #2: Yes

5. Review Comments to the Author

Reviewer #1: Your paper, "Parents’ views of psychological research with children: Barriers, benefits, personality,

and psychopathology" represents an interesting and necessary piece of research, which has important implications for the recruitment of young people into research studies. It is a well considered piece of research, which uses both questionnaire methods and psychometric analyses to develop a measure of parental barriers and benefits for participating in research.

However, I have some concerns and comments which I think would strengthen the paper prior to publication in PLOS One.

1. My first concern is around sample size of the questionnaire component. Can you provide justification that a sample size of 109 parents is sufficient for the analyses you conducted? For example, did you conduct a power calculation to determine a suitable target sample size? Is there statistical literature you could cite to support that this sample size is sufficient for Explorative SEM? You discuss this well at the start of your limitations section, however discussions of any power calculations may be useful at this earlier stage of the paper.

2. On page 7, line 160, you note an important research gap relating to a future focus on research with infants and school children. This should be foregrounded in the introduction section to the paper, as it comes as a bit of a surprise to read this for the first time amongst the study aims.

3. A minor point, but I think one that needs clarification - at several points in the paper you make reference to childrens' higher education (e.g., p.2 line 19; p.4, line 80). Do you mean parents' levels of higher education? Higher education doesn't typically begin until 18 years +, unless your definition of "children" is very broad in age range, this seems to be a little unclear.

4. In table 1, can you clarify that family income is per month?

5. A general observation, but I felt that much of the information given in your 'Measures' section was extraneous, and would be a better fit for others sections of the paper (introduction or discussion). As a result this section was overly lengthy to read, and could be much briefer and to the point.

6. A minor formatting point to be corrected later - but there is inconsistent in text referencing throughout the manuscript. This is sometimes names in parentheses, and sometime a numerical reference. e.g., "(Baker et al., 2011)" or "[51]".

7. On page 14, line 297 you state, "Since it was not possible to skip questions in the online study, there were no missing values in the data set." While good to have a statement relating to missing data, I wonder if you could consider the limitation of forced entry questions (where there is no option to skip). Did you check data for outliers / repeated item selection (e.g. participants selecting the same number on a scale for repeated questions), as it may be that participants provided meaningless or out of range responses simply to proceed.

8. Page 17, line 347, you state that same parents had participated in up to 200 studies. It would be helpful to see a discussion of self-selection bias in your limitations section. How many of your participants were at the high end of previous study participation? What was the mean level of participation in your sample? Were high responders likely to have (e.g.) higher levels of education (which may also contribute to your finding that parental education was negatively associated with barriers to participation.

The discussion section is comprehensive with good consideration of applied impact of the work. The limitations section is especially strong, and I was glad to see discussions of larger sample sizes, and CFA / Full SEM models to further validate the initial 'pilot' work conducted here.

Reviewer #2: This is a very interesting paper, and one that should be published. I have made some suggestions that should be addressed to ensire that paper is ready for publication.

Introduction

Whilst there is some mention of theory, the Health Belief Model, I think that the paper would benefit from more focus on theoretical underpinnings– so maybe one paragraph outlining one or more relevant theories.

The rationale needs to be strengthened and written more concisely. There also needs to be more explanation put forward for developing new measures.

Some typos - for e.g. pg. 3, line 48 – Compared to studies in adults.

Method

More information is needed in this section.

There needs to be a design section, which explains what the research set out to do– so cross sectional survey and also development of new measures.

In the school and kindergarten, was there a gatekeeper -more information needed here around the recruitment and consent process.

Also, what social platforms and how was consent gained?

Previous study participation and current willingness to participate sections need a clearer explanation.

Were the measures that were developed piloted? More information on the development of these measures would seem appropriate.

The statistical analysis section seems to only report around the development of the measures.

Results seem ok, but need to be organised clearer.

Discussion is quite long and would benefit from some reorganisation, headings and being made more concise.

There are typos throughout that need addressed.

6. PLOS authors have the option to publish the peer review history of their article (what does this mean?). If published, this will include your full peer review and any attached files.

Reviewer #1: No

Reviewer #2: No

---

## [Author Response · Author response to Decision Letter 0]

11 May 2023

Dear Dr. Letina,

Dear reviewer,

Thank you very much for your letter, and for the very helpful review of our manuscript titled "Parents’ views of psychological research with children: Barriers, benefits, personality, and psychopathology".

We thank you and the reviewer for the thoughtful and constructive feedback and suggestions for improving the manuscript. We have revised the manuscript according to your recommendations. We have carefully considered and responded in detail to each of the points made by you and the reviewer. Please find our actions detailed in the following. 

Another short note about the result part: In the version with marked changes, a lot is marked. This is because the order/structure was changed according to the recommendation of a reviewer (no substantive changes were made; except minimally as suggested by the reviewers).

1. We have ensured that the revised manuscript complies with the formal requirements, in particular naming of the files.

2. The original data have been uploaded to OSF and the DOI is mentioned in the revised manuscript on page 13: “The original data is available at: DOI: 10.17605/OSF.IO/JVFNA.”

3) We have changed the reference Bleichhardt & Hiller (unpublished) as recommended.

4. The Supporting Information has been cited in the revised manuscript as recommended by the journal.

Comments reviewer 1:

Reviewer #1: Your paper, "Parents’ views of psychological research with children: Barriers, benefits, personality, and psychopathology" represents an interesting and necessary piece of research, which has important implications for the recruitment of young people into research studies. It is a well considered piece of research, which uses both questionnaire methods and psychometric analyses to develop a measure of parental barriers and benefits for participating in research. However, I have some concerns and comments which I think would strengthen the paper prior to publication in PLOS One.

1. My first concern is around sample size of the questionnaire component. Can you provide justification that a sample size of 109 parents is sufficient for the analyses you conducted? For example, did you conduct a power calculation to determine a suitable target sample size? Is there statistical literature you could cite to support that this sample size is sufficient for Explorative SEM? You discuss this well at the start of your limitations section, however discussions of any power calculations may be useful at this earlier stage of the paper.

[AU]: Thank you for highlighting this point. We fully agree with you that we should mention the appropriate sample size already in the method. A power analysis showed an adequate sample size of N = 88 (G*Power, ρ = .30, α = 0.05, Power (1-β) = .90) for correlations and a minimum limit for Structural Equation Modeling (SEM) is recommended to be N = 100–150 (Ding, Velicer & Harlow, 1995; Tabachnick & Fidell, 2007), so that at least N = 100 participants were targeted (including 10% drop-outs approx. 110). We have now included these additions on page 8 of the revised manuscript.

“A power analysis showed an adequate sample size to be N = 88 (G*Power, ρ = .30, α = 0.05, Power (1-β) = .90) for correlations and a minimum limit for Structural Equation Modeling (SEM) is recommended to be N = 100–150 [40, 41]1, including 10% drop-outs, the target size was at least N = 110.” (on page 8)

1(Ding, Velicer & Harlow, 1995; Tabachnick & Fidell, 2007)

References added in the discussion section: “The sample size is clearly at lower bound for the computation of ESEM models [44, 45] and a replication in larger samples is definitely needed.” (on page 23)

2. On page 7, line 160, you note an important research gap relating to a future focus on research with infants and school children. This should be foregrounded in the introduction section to the paper, as it comes as a bit of a surprise to read this for the first time amongst the study aims.

[Au]: Thank you for the comment, which we find very understandable. We have now mentioned the research gap regarding age earlier in the revised version of the manuscript on page 4. In addition, we have changed the focus (especially in the section "The current study") to children and deleted the word "adolescents" in some places (on page 5).

“Overall, however, there is a research gap regarding (parental) factors for willingness to participate in more general psychological studies (basic research), especially among infants and school children from the general population (as opposed to specific topics and target groups such as children with physical illnesses).” (on page 4)

3. A minor point, but I think one that needs clarification - at several points in the paper you make reference to childrens' higher education (e.g., p.2 line 19; p.4, line 80). Do you mean parents' levels of higher education? Higher education doesn't typically begin until 18 years +, unless your definition of "children" is very broad in age range, this seems to be a little unclear.

[Au]: Yes, a very good point. Thank you very much. Data on educational attainment generally refer to parents, as in our study. We have presented this more precisely. However, in the abstract we have deleted this example out because it is ambiguous here and we can explain it better in the text.

e.g., “Previous studies have found that European participants or being White, higher education of parents and higher socio-economic status are overrepresented in child and adolescent research, while cultural minorities and participants with risk behaviors (e.g., substance use) are underrepresented [27–30].” (on page 4)

4. In table 1, can you clarify that family income is per month?

 [Au]: Thank you, we have added that.

5. A general observation, but I felt that much of the information given in your 'Measures' section was extraneous, and would be a better fit for others sections of the paper (introduction or discussion). As a result this section was overly lengthy to read, and could be much briefer and to the point.

[Au]: Thank you for your recommendation. In the revised version, we have shortened the measures section and have instead placed some information in the introduction (“The current study” section). For example, we omitted the first section under Measures, abridged the original information, and placed it under the "The current study" section. In addition to a more stringent presentation of P-BERQ and P-BARQ, we have made minor cuts to other measurement methods (e.g., BFI-10).

e.g. “The aims and hypotheses of this study were: (1) The development of standardized self-report measures for assessing parents’ perceived barriers and benefits of psychological studies with children. Two questionnaires (benefits/barriers) were developed based on the above empirical findings (e.g., short duration, individual feedback as perceived benefits) and on a similar questionnaire on barriers regarding own study participation from the adult population [42] and were validated in the present pilot study.” (on page 7)

e.g., “On the basis of the above findings and the BRPQ, 15 items were formulated (e.g., "I cannot find time to participate.", "I am thinking about what negative effects participation could have for my child.", "I do not want others to know personal information about our family."; all items see S1 Table).” (on pages 10 and 11)

6. A minor formatting point to be corrected later - but there is inconsistent in text referencing throughout the manuscript. This is sometimes names in parentheses, and sometime a numerical reference. e.g., "(Baker et al., 2011)" or "[51]".

[Au]: Thank you for your comment. We have now corrected this point and consistently used a reference style in number format.

7. On page 14, line 297 you state, "Since it was not possible to skip questions in the online study, there were no missing values in the data set." While good to have a statement relating to missing data, I wonder if you could consider the limitation of forced entry questions (where there is no option to skip). Did you check data for outliers / repeated item selection (e.g. participants selecting the same number on a scale for repeated questions), as it may be that participants provided meaningless or out of range responses simply to proceed.

[AU]: Thank you very much for your comment. We had checked for duplicates as well as consistent response patterns. One person (0.9%) always gave the same answer on the benefits questionnaire (14 items) and another person (0.9%) always gave the same answer on barriers (15 items), but no person across the two questionnaires (or all questionnaires), so we did not exclude any participant for this reason. We made an addition to this aspect in the revised version.

“We checked for duplicates and response patterns (one person responded the same to each of the items in the P-BARQ or P-BERQ, but no person responded the same across all questionnaires, so no exclusion occurred).” (on page 13)

8. Page 17, line 347, you state that same parents had participated in up to 200 studies. It would be helpful to see a discussion of self-selection bias in your limitations section. How many of your participants were at the high end of previous study participation? What was the mean level of participation in your sample? Were high responders likely to have (e.g.) higher levels of education (which may also contribute to your finding that parental education was negatively associated with barriers to participation.

[AU]: We agree that further information on previous parental participation is helpful (especially because of the wide range) and should be discussed. In the revised version, we have further specified the parents' information on previous participation and included it in the limitations. (No significant correlation was found between education and previous number of parents’ study participations: r=.10, p=.32).

“Parents and children did not differ significantly in the number of participations in previous psychological studies (t(108) = 1.96, p =. 053, d = 0.26), with parents showing a very wide range in responses (range: 0 – 200, M = 4.13, SD = 21.60; most frequent answer with 67% no previous study participation).” (on page 14)

“Although the range of previous parental participation was very large (0 – 200 previous study participation; M = 4.13, SD = 21.60), most parents had not previously participated in any psychological study (67% no previous study participation), so it can be speculated that possibly both parents who are more open-minded and more skeptical about studies participated in our survey.” (on page 24)

Comments reviewer :

Reviewer #2: This is a very interesting paper, and one that should be published. I have made some suggestions that should be addressed to ensire that paper is ready for publication.

1. Introduction: Whilst there is some mention of theory, the Health Belief Model, I think that the paper would benefit from more focus on theoretical underpinnings– so maybe one paragraph outlining one or more relevant theories.

[AU]: We thank you for this suggestion for improvement and agree that our manuscript can benefit greatly from a stronger theoretical foundation. Whether people participate in studies is related to decision making. In the revised version of the manuscript, we elaborate on prospect theory (Kahneman & Tversky, 1979) and heuristics (e.g., availability heuristic, Kahneman, 2011). The Health Belief Model (Rosenstock, 1974) is now presented in more detail in the revised version (on pages 4 and 5). Therefore, we deleted the single sentence on the Health Belief Model on page 4 and wrote a separate section as recommended.

“In the context of study participation, theories of decision-making and prediction of behavior are relevant. The Prospect Theory [34] assumes that people choose the behavior in which the subjectively expected benefit combined (multiplied) with the expected probability of this positive benefit is the highest. Heuristics may play a role in this process [35]; in the case of study participation, for example, the availability heuristic (e.g., memories of study advertisements or previous study participation). Because study participation is in a social and societal context, altruistic behavior (i.e., selfless behavior) is also assumed in study participation, which in explanatory approaches may be influenced by several factors, such as situational factors (e.g., mood, time pressure). A reference to health behaviors is provided by the Health Belief Model [36], which assumes that behavior depends in particular on a cost-benefit trade-off, i.e., on assumed advantages and disadvantages of the behavior. Thus, parents would be assumed to agree to and participate in a study if the benefits are perceived to be high and the barriers are perceived to be low [21, 24].” (on pages 4 and 5)

2. The rationale needs to be strengthened and written more concisely. There also needs to be more explanation put forward for developing new measures.

[AU]: Thank you for your helpful comment. In the revised version, we have made the introduction more stringent and clarified the rational. To do this, we deleted two sentences on page 3 (which interrupted the stringency), emphasized the research gap on page 4, and clarified the objectives and hypotheses on page 7.

“Overall, however, there is a research gap regarding (parental) factors for willingness to participate in more general psychological studies (basic research), especially among infants and school children from the general population (as opposed to specific topics and target groups such as e.g., children with physical illnesses).” (on page 4)

“The aims and hypotheses of this study were: (1) The development of standardized self-report measures for assessing parents’ perceived barriers and benefits of psychological studies with children. Two questionnaires (benefits/barriers) were developed based on the above empirical findings (e.g., short duration, individual feedback as perceived benefits) and on a similar questionnaire on barriers regarding own study participation from the adult population [42] and were validated in the present pilot study. Given that these were new developments, we expected at least satisfactory psychometric qualities (in terms of reliability, factorial, convergent, and discriminant validity).” (on page 7)

3. Some typos - for e.g. pg. 3, line 48 – Compared to studies in adults

[AU]: Thank you for pointing this out. We have carefully reviewed and corrected the manuscript.

4. Method: More information is needed in this section.

There needs to be a design section, which explains what the research set out to do– so cross sectional survey and also development of new measures.

In the school and kindergarten, was there a gatekeeper -more information needed here around the recruitment and consent process.

Also, what social platforms and how was consent gained?

Previous study participation and current willingness to participate sections need a clearer explanation.

Were the measures that were developed piloted? More information on the development of these measures would seem appropriate.

The statistical analysis section seems to only report around the development of the measures.

[AU]: Thank you for the follow-up questions and recommended additions for the method section. We agree that a design section is missing, so in the revised version we have added design to the Procedure section (Design and procedure, page 9) and provided more information about it.

Recruitment in kindergarten and schools was done via an e-mail distribution list. The e-mail included information on the nature, objectives, procedure and content of the study as well as the link to the study. When the link was opened, potential participants received detailed information and gave informed consent (by explicitly checking a box that the study information had been read and understood and that one was volunteering to participate in the study as opposed to the option of not reading/not understanding the study information or not volunteering to participate). I.e., no other people were involved in the recruitment (like some kind of gate keeper). We have now added this to the revised version. In addition to kindergartens and schools, recruitment took place via social platforms (on Facebook, Instagram, and Telegram) and on parent forums (where parents share problems, for example), where a flyer analogous to the e-mail was posted (informed consent as described above at the beginning of the online survey). We have now described this point more specifically in the revised version. Regarding the point of piloting/validation: the present study was a piloting of the questionnaires, in which they were validated for the first time. We have clarified this point in the introduction under "The current study". Also, we have already described in more detail the development of the questionnaires in the objectives. In the statistical analyses we have added the further procedures, i.e. correlation analyses and t-tests.

“Design and procedure

The present study was a cross-sectional online survey to pilot the newly developed questionnaires and investigate the above-mentioned relationships.” (on page 9)

“Recruitment for the survey titled "What do parents think about psychological studies with children" took place via a primary school and a kindergarten in a large German city as well as via social media (Facebook, Instagram, and Telegram) and parents' forums (e.g., on leisure or parenting). At school and kindergarten, parents received an e-mail with study information (type, content, and duration) and the link to the study (i.e., no other persons such as gatekeepers were involved in the recruitment). Recruitment in social media and on platforms for parents included a post with the same study information and the link to the study. Before participants could begin the online survey, they gave informed consent after receiving detailed study information (i.e., checking boxes indicating that information was read and understood and that they willingly participated in the study as opposed to the option of not read, not understood, or no voluntary participation). All participants then initially completed sociodemographic information and the questionnaires listed under Measures.” (on pages 9 and 10)

“The aims and hypotheses of this study were: (1) The development of standardized self-report measures for assessing parents’ perceived barriers and benefits of psychological studies with children. Two questionnaires (benefits/barriers) were developed based on the above empirical findings (e.g., short duration, individual feedback as perceived benefits) and on a similar questionnaire on barriers regarding own study participation from the adult population [42] and were validated in the present pilot study.” (on page 7)

“Pearson correlation coefficients were calculated to test the relationships to the second research question. In addition, t-tests were conducted to examine comparisons between children and parents in terms of estimated current willingness to participate and number of previous study participations.” (on page 13)

5. Results seem ok, but need to be organised clearer.

[AU]: Thank you for this suggestion for improvement. We can understand this comment very well. Originally, our approach was to first describe the factor structure in order to be able to describe the participant characteristics also in terms of the subscales of the new questionnaires (i.e., which items combined to which subscale). However, we agree that the structure of the results can be improved. In the revised version of the manuscript, we now first describe the "Participants’ characteristics" (here referring to P-BERQ/P-BARQ item level) and then the factor structure, reliability and validity of the new measurement instruments. We have also moved table 3 back a little, since the next step will deal with relationships according question 2.

6. Discussion is quite long and would benefit from some reorganisation, headings and being made more concise.

[AU]: Thank you for pointing this out; we have added subsections to the discussion and shortened it somewhat in the revised version (e.g. pages 20 and 21).

“P-BARQ and P-BERQ as newly developed questionnaires” (on page 19)

“Willingness to study participation and sociodemographic data” (on page 20)

“Willingness to study participation and parental personality traits and psychopathology” (on page 21)

“Implications and future studies” (on page 22)

7. There are typos throughout that need addressed.

[AU]: Thank you for reading carefully. We have checked the entire manuscript for typos and improved it.

Yours sincerely,

The authors

---

## [Decision Letter · Decision Letter 1]

5 Jun 2023

Parents’ views of psychological research with children: Barriers, benefits, personality, and psychopathology

PONE-D-23-02403R1

Dear Dr. Jungmann,

We’re pleased to inform you that your manuscript has been judged scientifically suitable for publication and will be formally accepted for publication once it meets all outstanding technical requirements.

Kind regards,

Srebrenka Letina, Ph.D.

Academic Editor

PLOS ONE

Additional Editor Comments (optional):

Reviewers' comments:

Reviewer's Responses to Questions

**Comments to the Author**

1. If the authors have adequately addressed your comments raised in a previous round of review and you feel that this manuscript is now acceptable for publication, you may indicate that here to bypass the “Comments to the Author” section, enter your conflict of interest statement in the “Confidential to Editor” section, and submit your "Accept" recommendation.

Reviewer #1: All comments have been addressed

Reviewer #2: All comments have been addressed

2. Is the manuscript technically sound, and do the data support the conclusions?

Reviewer #1: Yes

Reviewer #2: Yes

3. Has the statistical analysis been performed appropriately and rigorously? 

Reviewer #1: Yes

Reviewer #2: Yes

4. Have the authors made all data underlying the findings in their manuscript fully available?

Reviewer #1: Yes

Reviewer #2: Yes

5. Is the manuscript presented in an intelligible fashion and written in standard English?

Reviewer #1: Yes

Reviewer #2: Yes

6. Review Comments to the Author

Reviewer #1: Thank you for responding well to my comments. You've taken the time to consider these, and reply constructively and meaningfully. Thank you for your efforts on this.

I'm happier with the manuscript, and happy to recommend for publication.

Reviewer #2: I am happy with the amendments that the authors have carried out. The authors have addressed each comment/suggestion fully and the paper is now in good shape for publication.

7. PLOS authors have the option to publish the peer review history of their article (what does this mean?). If published, this will include your full peer review and any attached files.

Reviewer #1: No

Reviewer #2: No

---

## [Editor Report · Acceptance letter]

14 Jun 2023

PONE-D-23-02403R1 

Parents’ views of psychological research with children: Barriers, benefits, personality, and psychopathology 

Dear Dr. Jungmann:

I'm pleased to inform you that your manuscript has been deemed suitable for publication in PLOS ONE. Congratulations! Your manuscript is now with our production department. 

Kind regards, 

on behalf of

Dr. Srebrenka Letina 

Academic Editor

PLOS ONE